# New Data on Comparative Cytogenetics of the Mouse-Like Hamsters (*Calomyscus* Thomas, 1905) from Iran and Turkmenistan

**DOI:** 10.3390/genes12070964

**Published:** 2021-06-24

**Authors:** Svetlana A. Romanenko, Vladimir G. Malikov, Ahmad Mahmoudi, Feodor N. Golenishchev, Natalya A. Lemskaya, Jorge C. Pereira, Vladimir A. Trifonov, Natalia A. Serdyukova, Malcolm A. Ferguson-Smith, Mansour Aliabadian, Alexander S. Graphodatsky

**Affiliations:** 1Institute of Molecular and Cellular Biology (IMCB), Siberian Branch of Russian Academy of Sciences (SB RAS), 630090 Novosibirsk, Russia; lemnat@mcb.nsc.ru (N.A.L.); vlad@mcb.nsc.ru (V.A.T.); serd@mcb.nsc.ru (N.A.S.); graf@mcb.nsc.ru (A.S.G.); 2Zoological Institute (ZIN), Russian Academy of Sciences (RAS), 199034 Saint-Petersburg, Russia; malikovzin@mail.ru (V.G.M.); f_gol@mail.ru (F.N.G.); 3Department of Biology, Faculty of Science, Urmia University, Urmia 5756151818, Iran; ahmad.mahmoodi1985@gmail.com; 4Animal and Veterinary Research Centre (CECAV), University of Trás-os-Montes and Alto Douro (UTAD), 5000-801 Vila Real, Portugal; jorgecpereira599@gmail.com; 5Cambridge Resource Centre for Comparative Genomics, Department of Veterinary Medicine, University of Cambridge, Cambridge CB3 OES, UK; maf12@cam.ac.uk; 6Department of Natural Science, Novosibirsk State University, 630090 Novosibirsk, Russia; 7Department of Biology, Faculty of Sciences, Ferdowsi University of Mashhad, Mashhad 9177948974, Iran; aliabadi@um.ac.ir

**Keywords:** banding, chromosome painting, fluorescent in situ hybridization, karyotype, molecular cytogenetics, painting probes, type locality

## Abstract

The taxonomy of the genus *Calomyscus* remains controversial. According to the latest systematics the genus includes eight species with great karyotypic variation. Here, we studied karyotypes of 14 *Calomyscus* individuals from different regions of Iran and Turkmenistan using a new set of chromosome painting probes from a *Calomyscus* sp. male (2n = 46, XY; Shahr-e-Kord-Soreshjan-Cheshme Maiak Province). We showed the retention of large syntenic blocks in karyotypes of individuals with identical chromosome numbers. The only rearrangement (fusion 2/21) differentiated *Calomyscus elburzensis, Calomyscus mystax mystax*, and *Calomyscus* sp. from Isfahan Province with 2n = 44 from karyotypes of *C. bailwardi, Calomyscus* sp. from Shahr-e-Kord, Chahar Mahal and Bakhtiari-Aloni, and Khuzestan-Izeh Provinces with 2n = 46. The individuals from Shahdad tunnel, Kerman Province with 2n = 51–52 demonstrated non-centric fissions of chromosomes 4, 5, and 6 of the 46-chromosomal form with the formation of separate small acrocentrics. A heteromorphic pair of chromosomes in a specimen with 2n = 51 resulted from a fusion of two autosomes. C-banding and chromomycin A3-DAPI staining after G-banding showed extensive heterochromatin variation between individuals.

## 1. Introduction

The mouse-like hamsters (genus *Calomyscus* Thomas, 1905) are distributed mosaically and strictly associated with the rocky habitats of South-Western Syria, Iran, Afghanistan, Western Pakistan, Azerbaijan (Nakhichevan), and Turkmenistan. Initially, the genus was included in the subfamily Cricetinae of the family Cricetidae [1,2,3,4,5], but later, according to the results of morphological [6] and comparative molecular-genetic analyses [7], the genus was considered as a taxon of a monotypic family, Calomyscidae Vorontsov and Potapova, 1979, characterized by brush-tailed mouse-like appearance [8].

The taxonomic structure of the genus *Calomyscus* is still highly questionable. The type species of the genus, *C. bailwardi*, Thomas, 1905, was described from the vicinity of Mala-Imir (modern Izeh), Khuzestan Province, the South-West of Iran. Then, Thomas [9] discerned *C. hotsoni* Thomas, 1920 and *C. baluchi* Thomas, 1920 from Pakistan based on mild morphometric and external differences. Later, *C. mystax* Kaschkarov, 1925 from Bolshoy Balkhan Mountain, Turkmenistan was described [10]. Ognev and Heptner considered *C. mystax* to be synonymous with *C. hotsoni* [11]. According to their view, the genus comprises only two species: *C. bailwardi* (including two subspecies—*C. bailwardi bailwardi* and *Calomyscus*
*bailwardi baluchi*) and *C. hotsoni* [11]. In addition, the authors did not exclude that the differences between those forms could be less than interspecific. Argyropulo considered those specific names to be synonymous with *C. bailwardi* [12]. Later, *Calomyscus* was discovered in the South-East of Transcaucasia (Nakhichevan) and considered *C. bailwardi* [13]. In the same year, *C. elburzensis* Goodwin, 1939 was described from the Central Elburz (Iran). This author did not exclude *C. elburzensis* and/or *C. mystax* from being a subspecies of *C. hotsoni* [14]. Then, for a long time the genus *Calomyscus* was considered to be monotypic [15,16,17,18,19], including all its forms as subspecies. In that period, *C. bailwardi mustersi* Ellerman, 1948 (from the vicinity of Kabul, Afghanistan) and *C. grandis* Schlitter et Setzer, 1977 (from the vicinity of Fasham, the southern slopes of Elburz) were described [20]. *Calomyscus tsolovi* Peshev, 1991 from the southwest of Syria [21,22] was the latest species of the genus to be described, distinguished on the basis of only external characters. It was considered that *C. hotsoni* is a subspecies of *C. bailwardi*, which included all the forms from Iran [22].

The first data on the cytogenetics of the mouse-like hamsters were obtained by Matthey, who described the karyotype (diploid number 2n = 32) of a specimen from the vicinity of Julfa (Nakhichevan, Azerbajan) [23]. Then, the form with a diploid number of 2n = 30 was found in the southwest of Turkmenistan. On the basis of these data the authors described the Transcaucasian form as a new species *C. urartensis* Vorontzov et Kartavtseva, 1979 and restored the specific state of *C. mystax* which, according to them, included all the forms from Turkmenistan together with *C. elburzensis* [24]. In the same publication the hypothetical composition of the genus was given: (1) *C. urartensis*—Azerbaijan: Nakhichevan (the type locality) and the adjacent territory of Iran; (2) *C. mystax*—Turkmenistan: Bolshoy Balkhan Mountain (the type locality) and Mali Balkhan, the Turkmenian Kopet-Dag; Iran: mountains of the North Khorasan and the eastern part of Elburz up to the vicinity of Tehran; (3) *C*. *bailwardi*—Iran: Khuzestan, vicinity of Izeh (the type locality), Zagros Mountains in the Fars Province and the Zard-Kuh Mountains in Isfaghan Province; (4) *C*. *hotsoni*—Pakistan: The Central Makran Chain, the region of Panjgur (the type locality); (5) *C. baluchi*, including two subspecies: (a) *C. baluchi baluchi*—Pakistan, vicinity of Kelat (the type locality); (b) *C. baluchi mustersi*—Afghanistan, vicinity of Kabul (the type locality).

Later, extensive karyotype variations were shown for the mouse-like hamsters from different regions [25]. The variation concerned not only diploid chromosome numbers, but the fundamental number of autosomal arms (FNa) and the amount and distribution of heterochromatin. As a result of comparative karyology and experimental hybridization of different chromosomal forms from the former USSR, *C. urartensis* and *C. mystax* were recognized as different species. The nominative subspecies *C. mystax mystax* (diploid chromosome number 2n = 44, FNa = 46) was established and two new forms from Turkmenistan were described—*C. m. zykovi* Meyer et Malikov, 2000 (diploid number 2n = 30), which was for the first time karyotyped by Vorontzov et al. [24], and *C. firiuzaensis* Meyer et Malikov, 2000 (diploid number 2n = 44, FNa = 58), which was defined by Graphodatsky et al. [25] as karyotype 3. The “2n = 44” karyotypes of *C. m. mystax* and *C. firiuzaensis* differed from each other in the patterns of C- and G-banding. The interspecific hybrid F1 males were sterile, while the hybrids of both sexes between *C. m. mystax* and *C. m. zykovi* were fertile [26]. Nevertheless, only eight species of the genus—*C. bailwardi, C. hotsoni, C. baluchi, C. mystax, C. elburzensis, C. grandis, C. urartensis,* and *C. tsolovi*—were included in the checklist [8].

The first multivariate craniometric analysis of *Calomyscus* demonstrated distinct clusters [27,28] that did not always correspond to the karyotypic forms. For instance, *C. m. mystax* and *C. m. zykovi,* in spite of their karyotype differentiation, were lumped in the same craniometric sub-cluster. A large form from the vicinity of Tehran (presumably *C. grandis*), having the same karyotype as *C. m. mystax* [25] occurred to be apart from all the forms analyzed (unfortunately, previous karyotype comparisons were made using conventional banding techniques only). However, the craniometrics [29] and phylogenetic trees based on *CytB* [30] of the forms were in concordance. At the same time, the karyotyped *C. firiuzaensis* occurred in the same craniometric sub-cluster as the non-karyotyped *C. elburzensis* from the type locality and populations of northeastern Iran (Khorossan Province) and northwestern Afghanistan (Gerat Province), which strongly suggests them to be conspecific. However, recent molecular evidence distinguishes seven species (*C. bailwardi, C. hotsoni, C. baluchi, C. mystax, C. elburzensis, C. grandis,* and *C. urartensis*) and three additional lineages from the Zagros Mountains [31].

Here, we present a detailed karyotype description of 14 specimens of *Calomyscus* spp. from five different localities in Iran, including the type localities of *C. bailwardi* and *C. elburzensis,* and *C. m. mystax* from Turkmenistan.

## 2. Materials and Methods

### 2.1. Compliance with Ethical Standards

All applicable international, national, and/or institutional guidelines for the care and use of animals were followed. All experiments were approved by the Ethical committee at the ZIN RAS, Russia (permission No. 2-7/02-04-2021 of 2 April 2021) and the Ethics Committee on Animal and Human Research at the IMCB SB RAS, Russia (protocol No. 01/19 of 21 January 2019).

### 2.2. Species Sampled

The live individuals captured in Iran in 1998, 2014, and 2016 field seasons became the ancestors of two laboratory colonies—in the Department of Theriology of the Zoological Institute, St. Petersburg, and in Moscow Zoo. Here, we analyzed three individuals from the laboratory colony of Moscow Zoo (CSP1m, CSP2m, CSP3m) and two captive-born specimens form the Zoological Institute‘s colony (CMYS2m, CELB3m) (Table 1). Nine specimens examined were collected from free-living populations at five localities in Iran (Table 1 and Figure 1). It must be emphasized that in fact, we karyotyped natural individuals of all forms described so far, and the founder animals from the laboratory colonies were also analyzed karyologically. We consistently demonstrated stability of the karyotypes within the colonies, thus they represent a natural complexity.

### 2.3. Chromosome Preparation and Chromosome Staining

Cell lines and chromosome suspensions were obtained in the Laboratory of Animal Cytogenetics, IMCB SB RAS, Russia. The fibroblast cell lines were derived from lung, breastbone, and tail biopsies using enzymatic treatment (with trypsin, collagenase, and hyaluronidase) of tissues as described previously [32,33]. All the cell lines were stored in the IMCB SB RAS cell bank (“The general collection of cell cultures”, No. 0310-2016-0002).

Metaphase chromosome spreads were prepared as described previously [34]. We used colcemid to arrest the cell cycle in metaphase and ethidium bromide to directly inhibits chromosome condensation. C-banding using barium hydroxide octahydrate was made as described earlier [35]. G-banding was performed on chromosomes of all species prior to FISH by the standard trypsin/Giemsa procedure [36]. Chromomycin A3-DAPI-after G-banding (CDAG) was carried out following a previously published technique [37]. In short, after G-banding slides were heat denatured in the presence of formamide with consecutive fluorochrome staining.

### 2.4. Fluorescent in Situ Hybridization (FISH)

*Calomyscus* Cot-6 DNA extraction and FISH with the Cot DNA was performed following previously published protocols [38]. The set of chromosome-specific *Calomyscus* sp. (CSP17m) male probes was generated in the Cambridge Resource Centre for Comparative Genomics by DOP-PCR amplification of flow sorted chromosomes. Probes were labeled with biotin and digoxigenin by DOP-PCR amplification as described previously [39,40,41,42] (Figure 2).

### 2.5. Microdissection, Probe Amplification, and Labeling

Glass needle-based microdissection of small autosomes of *Calomyscus* sp. 3m (CSP3m) was performed on G-banded chromosomes as described [43]. One copy of each chromosome was collected. Chromosomal DNA was amplified and labeled using WGA kits (Sigma-Aldrich, Saint Louis, MO, USA).

### 2.6. Image Acquisition and Processing

Images were captured using the VideoTest-FISH software (VideoTesT) with a CCD camera (JenOptic) mounted on an Olympus BX53 microscope. Hybridization signals were assigned to specific chromosome regions identified by means of G-banding patterns photographed by the CCD camera. All images were processed in Corel Paint Shop Photo Pro X3 (Corel Corporation).

## 3. Results

### 3.1. Karyotype Descriptions

The *C. bailwardi* (CBAI1f) and *Calomyscus* sp. 21m (CSP21m) females from Khuzestan-Izeh, the specimens CSP17m, CSP20m, and CSP22f from Shahr-e-Kord, and the females *Calomyscus* sp. (CSP18f) and CSP23f from Chahar Mahal and Bakhtiari-Aloni have 2n = 46 (Figure 3). C-banding detected small heterochromatic blocks in pericentromeric regions of all chromosomes and a small interstitial heterochromatic block in the distal region of acrocentric X chromosomes (Figure 3). The block on X chromosome is AT-rich (see CDAG description below). A heteromorphic pair of chromosomes 4 composed of an acrocentric and a submetacentric was detected in karyotype of CBAI1f, CSP18f, CSP20m, CSP21m, and CSP22f, so FNa = 45. The individuals CSP17m and CSP23f have FNa = 44. The Y chromosome of CSP17m (metacentric) has a heterochromatic short arm (p-arm).

The karyotype of the male *Calomyscus* sp. 2m (CSP2m) was described and published previously [45]. It had 2n = 52 and FNa = 56. A similar karyotype structure is revealed for CSP1m with 2n = 52, FNa = 56 (Figure 4). The only difference between these two karyotypes is the heteromorphism of chromosome 1 due to heterochromatin variation. *Calomyscus* sp. 3m (CSP3m) has 2n = 51, FNa = 57, and carries a heteromorphic pair of chromosomes 1 consisting of a large submetacentric chromosome and two acrocentrics homologous to its short (p-) and long (q-) arms, respectively (Figure 4). C-banding reveals a large block of heterochromatin in the pericentromeric part of the biggest submetacentric chromosome (Figure 4).

The karyotypes of *C. elburzensis* individuals (CELB1f and CELB3m), *C. mystax mystax,* and *Calomyscus* sp. 26m (CSP26m) have 2n = 44 (Figure 5). As CELB1f carries a heteromorphic chromosome 11 (Figure 5a), the number of autosome arms differ in the two individuals examined: FNa = 61 for CELB1f and FNa = 62 for CELB3m. C-banding reveals additional heterochromatic arms in autosomes 1–9, X chromosomes, and the submetacentric homolog of chromosome 11 of *C. elburzensis* (Figure 5). G- and C-banded karyotypes of CELB3m were presented previously [47]. The *C. m. mystax* karyotype shows acrocentric chromosomes carrying heterochromatic blocks in pericentromeric regions (Figure 5b). The karyotype of CSP26m has FNa = 67. Heterochromatic blocks were revealed in pericentromeric regions of all chromosomes forming additional p-arms in pairs 1–13 (Figure 5c).

### 3.2. The Results of Chromomycin A3-DAPI-after G-banding (CDAG)

Pericentromeric regions of CSP1m, CSP3m, and CBAI1f are AT-rich (Figure 6a–c). There is no clear AT-rich pericentromeric heterochromatin in CSP17m and CSP18f (Figure 6d,e). The interstitial heterochromatic blocks on the X chromosomes of CSP17m, CSP18f, and CBAI1f are AT-rich (Figure 6c–e). Although the p-arms of autosomes are uniformly stained with chromomycin A3 in CSP26m and both CELB individuals, p-arms of their X chromosomes show alternation of AT- and GC-rich heterochromatin (Figure 6f–i); pericentromeric regions in these individuals are AT-rich. The pericentromeric regions of CMYS2m chromosomes are both GC- and AT-enriched with higher GC-content (Figure 6g). An AT-rich area is observed in the distal region of the p-arms of the X chromosomes. Weak staining with chromomycin A3 is characteristic of pericentromeric, interstitial, and distal regions of some chromosomes in all individuals. The Y chromosomes of all individuals are predominantly AT-rich.

### 3.3. The Flow Karyotype of CSP17m

The male CSP17m has a diploid number of 46 (Figure 3a). The chromosome complement of CSP17m is resolved into 21 separate peaks by flow cytometry (Figure 2). The chromosomal content of each peak was identified by hybridizing the probes derived from each peak onto G-banded chromosomes of the source specimen (Figure 7). Fifteen single chromosome-specific painting probes were obtained (1–9, 16, 17, 21, 22, X, and Y). Two probes each paint two pairs of autosomes (17 + 18, 19 + 20) and four peaks contain a mixture of three autosomes (10 + 12 + 19, 11 + 12 + 13, 11 + 13 + 14, and 15 + 16 + 18). We need to stress that the standard karyotype is not arranged in order of chromosome size as revealed by the flow karyotype. The presence of chromosomes 11, 12, and 13 in two distinct peaks can be explained by slight heteromorphism of the homologs detected by cytogenetic analysis of G-banded chromosomes. In the case of chromosome 19 the possible description is more complicated. A strong signal on chromosome 19 is revealed when we localized the painting probe (without Cot-6) derived from the peak containing chromosomes 10 + 12 (Figure 7e). The intensity of signals is much weaker when we made FISH with Cot-6 DNA, which indicates that this is due to cross hybridization of the repetitive elements of heterochromatin. FISH with the probe containing chromosomes 19 + 20 produced slight background signals on chromosomes 10 and 12. We suggest that microsatellite repeats could give such a signal distribution.

Microdissection-derived probes of small autosomes 20–25 of CSP3m were made (Figure 4b). The chromosomal content of each probe was identified by hybridizing these probes onto G-banded chromosomes of the CSP17m source specimen (Figure 7). The localization of these probes on CSP17m chromosomes identified the corresponding regions (Figure 3a and Figure 4b).

According to the results of the localization of painting-probes, chromosome painting using CSP17m paints show that hamster karyotypes with the same diploid chromosome numbers do not differ from each other. Regarding the 46-chromosome forms (CBAI1f, CSP17m, CSP18f, CSP20m, CSP21m, CSP22f, and CSP23f), individuals with 2n = 44 (CELB1f, CELB3m, CMYS2m, and CSP26m) showed fusion of 2/21, and individuals with 2n = 51–52 (CSP1m, CSP2m, CSP3m) demonstrated fissions of chromosomes 4, 5, and 6 with the formation of separate small acrocentrics. The heteromorphic pair of chromosomes in CSP3m was formed by the fusion of autosomes 1 and 6 of CSP17m.

## 4. Discussion

The mouse-like hamsters of the genus *Calomyscus* are widely spread but the recognition of species is complicated due to the morphological similarity of individuals, the possible intraspecific chromosomal variation and the occurrence of natural hybrids [25]. Therefore, the taxonomic analysis by morphometric structure and molecular cytogenetics of individuals from the localities of the named forms is particularly important.

The specimens, presumably belonging to “true” *C. bailwardi,* were initially recorded only from the Zagros mountains in west Iran [8]. At the same time, a high chromosomal variability of *Calomyscus* samples from different parts of the Zagros mountains (with diploid numbers varying from 37 to 52) [25,49] and molecular data [31], suggest that Zagros is colonized by more than one species of the genus. Musser and Carleton [8] stressed that Graphodatsky et al. [25] had karyotyped individuals from the *C. bailwardi* type locality (it seems that the authors meant karyotype 7 with 2n = 50 from the region of Persepolis). Moreover, the karyotype of another sample with unknown origin identified as *C. bailwardi* (2n = 44) was published previously [50].

Two specimens, *C. bailwardi,* CBAI1f, and CSP21m both from Izeh (type locality), have cytogenetically completely identical autosomal sets and X chromosomes (Figure 3b). The similarity of karyotypes and the geographical origin allow us, with a certain degree of caution, to classify CSP21m as *C. bailwardi*. The karyotype of both individuals differ from the karyotype 7 published by Graphodatsky et al. [25] by a lower diploid chromosome number (46 and 50, respectively) and chromosome morphology. The karyotypes also differ from the *C. bailwardi* karyotype presented by Radjabli et al. [50] not only by the diploid chromosome number and chromosome morphology, but also by the amount and distribution of heterochromatin. It is possible that the differences between the *C. bailwardi* karyotypes described here and presented by Radjabli et al. [50] can be explained by incorrect species identification in the previously published case. However, the high variability of chromosomal characteristics may indicate the need for additional karyological and molecular studies to determine the systematic status and composition of *C. bailwardi*.

The karyotypes of individuals with 2n = 46 from different habitats (CBAI1f and CSP21m from Khuzestan-Izeh; CSP17m, CSP20m, and CSP22f from Shahr-e-Kord; CSP18 and CSP23f from Chahar Mahal and Bakhtiari-Aloni) did not actually differ from each other (Figure 3). The only exception is that the specimens from Khuzestan-Izeh displayed clearer pericentromeric blocks of heterochromatin. Interestingly, the only analyzed individual from closely located Isfahan, CSP26m, has a different karyotype, characterized by a smaller diploid number and the presence of pronounced heterochromatic p-arms (Figure 5c).

Three individuals studied here from Shahdad tunnel, Kerman Province, have similar karyotypes with 2n = 51–52 (Figure 4). The slight differences are caused by variations in heterochromatin. The karyotype structure and heterochromatic composition of these individuals is similar to karyotype 6, described earlier by Graphodatsky et al. [25] in a female specimen from Kerman Province, Iran. All that can be assumed is the presence of larger blocks of heterochromatin in the pericentromeric regions of all chromosomes in the individual described earlier by Graphodatsky et al. [25].

The karyotypes of both *C. elburzensis* (CELB1f was captured in their type locality in Iran) studied here have 2n = 44 (Figure 5a). The karyotypes show good correspondence with the one published by Radjabli et al. [50] and with the karyotype 3 by Graphodatsky et al. [25], both by C- and G-banding. Nevertheless, in the article by Graphodatsky et al. [25], chromosomes 9 and 11 pairs are designated acrocentric, consistent with FNa = 58. Although, according to their C-banding figure, it can be seen that the pairing of chromosome 9 is submetacentric, which makes FNa = 60. In our case, we find FNa = 61 for CELB1f and FNa = 62 for CELB3m due to heteromorphism of chromosome 11. The obvious disagreement in chromosome morphology between C- and G-banded karyotypes (e.g., pair 7 is acrocentric in G-banded and submetacentric in C-banded karyotypes) can be explained by possible intraspecific variability in heterochromatin amount and distribution [50]. Considering the similarity of *C. elburzensis* type karyotype to karyotype 3, *C. firiuzaensis* should be a junior synonym of *C. elburzensis* taxonomically. This conclusion is fully consistent with the results of multivariate craniometric analysis [27,28]. In addition, despite the different level of resolution, it seems that the karyotypes of CELB1f and CELB3m are similar to the karyotype of *C. elburzensis* with FNa = 62 described by Shahabi et al. [49], despite their different collection sites.

Previously it was proposed, based on cytogenetic analysis with different banding techniques, that centric and tandem fusions and heterochromatin variations play a major role in the karyotype evolution of *Calomyscus* [25]. The use of chromosome specific painting probes allows us to support the hypothesis of high rates of chromosomal transformations by translocation and variation of heterochromatin. Karyotypes of *C. elburzensis, C. m. mystax,* and CSP26m differ from other karyotypes studied here by the only tandem fusion 2/21 (Figure 5). This fact is especially interesting because the respective individuals were caught in places distant from one another (Figure 1). Moreover, the karyotypes characterized by a large amount of heterochromatin have additional chromosome arms. The description of *C. m. mystax* karyotype indicates a correspondence to the karyotype of this species described earlier by Graphodatsky et al. [25].

All specimens studied here have stable karyotypes. Despite this, we reveal heteromorphic chromosome pairs in karyotypes of CSP2m (pair 1), CBAI1f, CSP18f, CSP20m, CSP21m, CSP22f (pair 4) and CELB1f (pair 11) (Figure 3 and Figure 5a). However, heteromorphism caused by different chromosomal rearrangements is quite often observed, especially among relatively young and fast-evolving species [51,52]. Heteromorphism in pairs of chromosomes 1 in CSP1m and 11 in CELB1f is caused by an additional heterochromatic arm on one of the homologs. The identification of the type of chromosomal rearrangement in pair 4 of CBAI1f and CSP18f is complicated. The different morphology of homologs could be the result of either pericentromeric inversion or centromere shift and additional investigations are needed. Moreover, the presence of some chromosomes in different peaks (e.g., 11, 12, and 13) of the flow sorted karyotype probably indicates heterochromatin variation in homologs or tiny chromosomal rearrangements (Figure 2).

## 5. Conclusions

Karyotype analysis plays an important role in the identification of morphologically similar species. Some chromosomal characteristics suggest the high plasticity of the *Calomyscus* genome and/or currently ongoing speciation process. However, neither similarity nor differentiation between the forms of *Calomyscus* in karyotype or molecular genetic markers can serve as an unambiguous indicator of their species rank. The results of morphometric and molecular genetic analysis of the diversity of *Calomyscus* [28,30,53,54,55], in combination with the previously published data, show that the genus demonstrates a lack of correlation between its karyotypic and morphometric structure and the reproductive incompatibility of the various forms. More integrative studies are required for an improved understanding of speciation in this genus, among which the data on reproductive isolation between genetically marked forms will be the most instructive.

## Figures and Tables

**Figure 1 genes-12-00964-f001:**
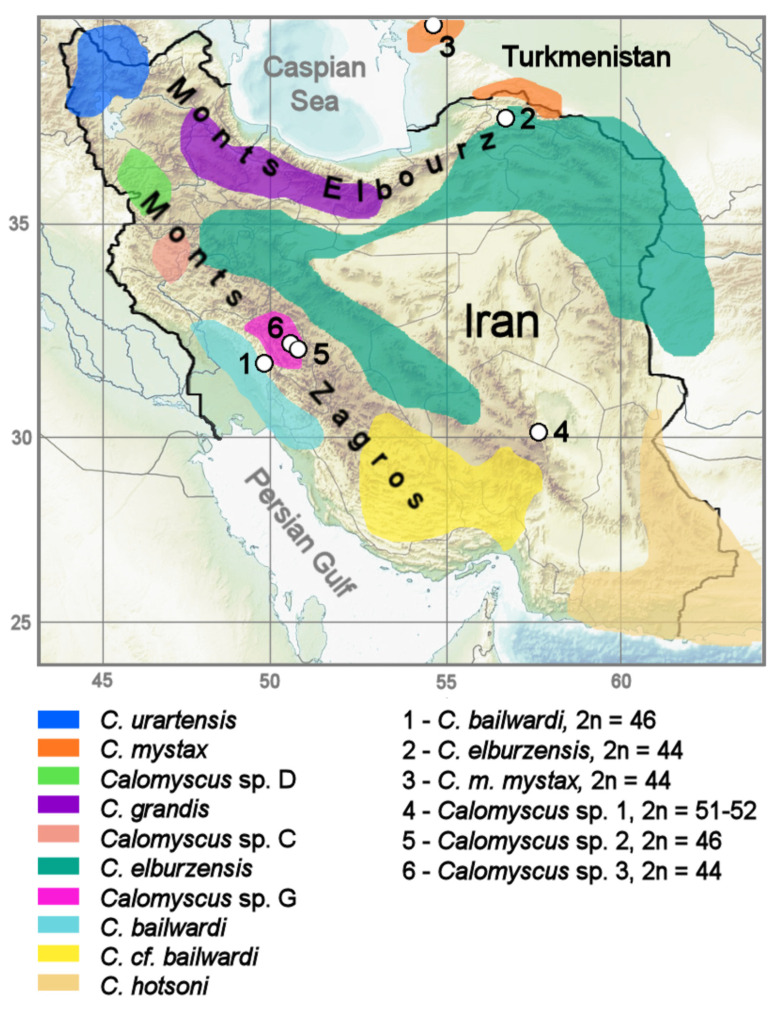
Geographic position of the localities studied: 1—Izeh, Khuzestan Province, Iran, 2—Korkhout Mt., North Khorasan Province, Bojnourd, Iran, 3—Maly Balkan Gershi (Arlandag) Mt., Bolshoi Balkhan, Turkmenistan, 4—Shahdad tunnel, Kerman Province, Iran, 5—Aloni and Cheshme Maiak, Shahr-e-Kord—Soreshjan, Chahar Mahal and Bakhtiaria Province, Iran, 6—southeastern ridge Zard-Kuh Mts., Isfahan Province, Iran. For detailed localities description see Table 1. *Calomyscus* distributional ranges in Iran and Turkmenistan are given in accordance with the mitochondrial DNA analyses [31].

**Figure 2 genes-12-00964-f002:**
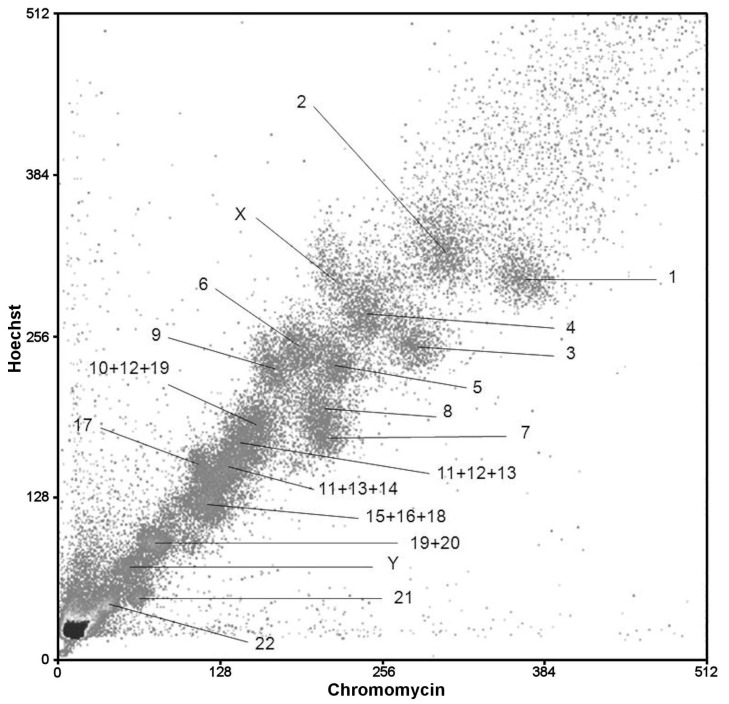
Bivariate flow karyotype of *Calomyscus* sp. (CSP17m) cell line with chromosomal assignments.

**Figure 3 genes-12-00964-f003:**
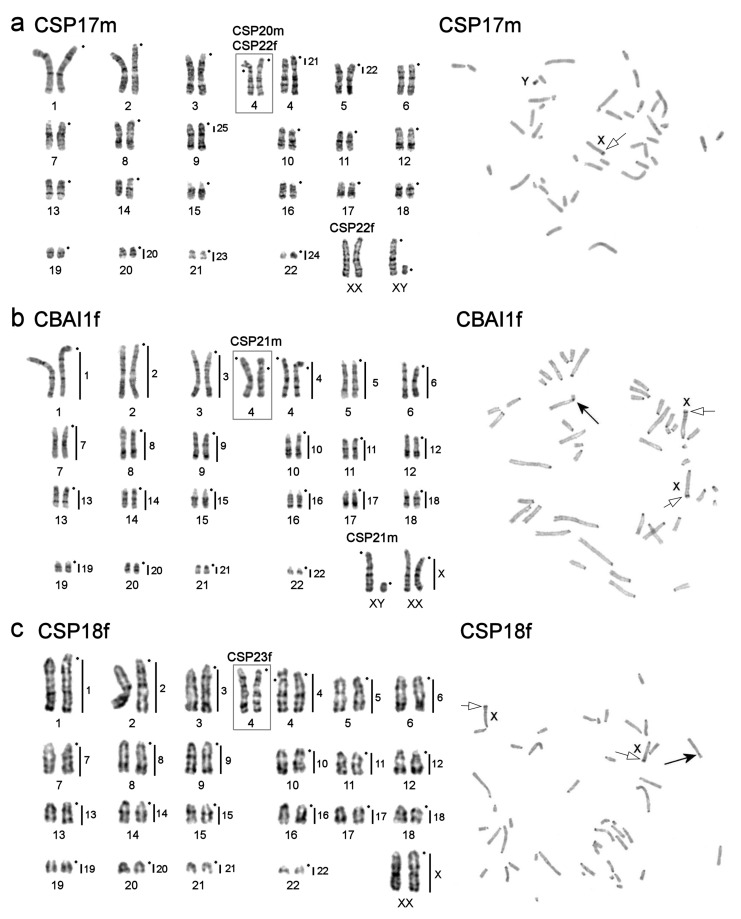
G-banded karyotypes (on the left) and C-banded metaphases (on the right) of the studied specimens with 2n = 46 and FNa = 44–45: (**a**) *Calomyscus* sp. 17m, *Calomyscus* sp. 20m (CSP20m), and *Calomyscus* sp. 22f (CSP22f), vertical black lines and figures on the right of G-banded chromosomes show localization of CSP3m probes; (**b**) *C. bailwardi* (the karyotype was presented previously in [44]) and *Calomyscus* sp. 21m (CSP21m); (**c**) *Calomyscus* sp. 18f and *Calomyscus* sp. 23f (CSP23f). In part b and c vertical black lines and figures on the right of G-banded chromosomes show localization of CSP17m probes. Black dots mark the position of centromeres. In C-banding photos the black arrows mark the submetacentric homolog of chromosome 4 and sex chromosomes (X and Y) are shown. The white arrows with black stroke mark the interstitial heterochromatic block in the distal region of X chromosomes.

**Figure 4 genes-12-00964-f004:**
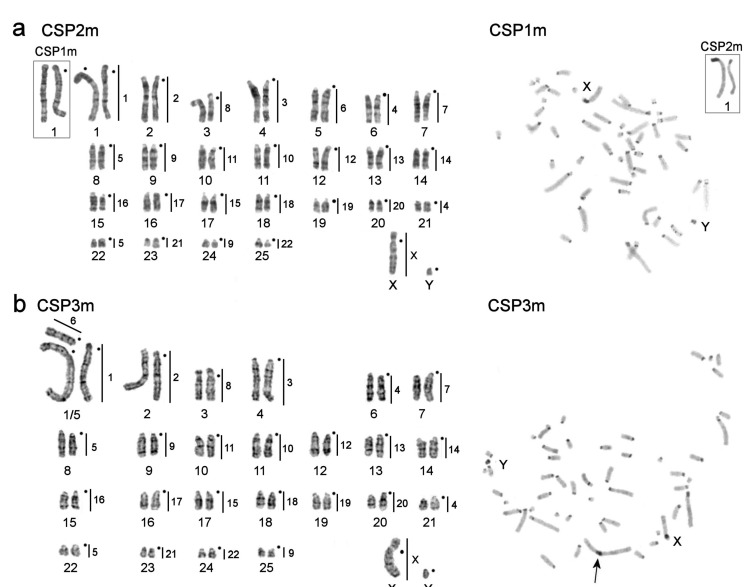
G-banded karyotypes (on the left) and C-banded metaphases (on the right) of the studied specimens with 2n = 51–52 and FNa = 56–57: (**a**) *Calomyscus* sp. 1m (CSP1m) and *Calomyscus* sp. 2m (CSP2m, corrected karyotype from Romanenko et al. [45], published as “*Calomyscus* sp. 1” in [46]) with 2n = 52, FNa = 5 6; (**b**) *Calomyscus* sp. 3m (CSP3m) with 2n = 51, FNa = 57*,* vertical black lines and figures on the right of G-banded chromosomes show localization of *Calomyscus* sp. 17m (CSP17m) probes. Black dots mark the position of centromeres. In C-banding photos the black arrow indicates a pericentromeric heterochromatin segment in the largest chromosome and sex chromosomes (X and Y) are shown.

**Figure 5 genes-12-00964-f005:**
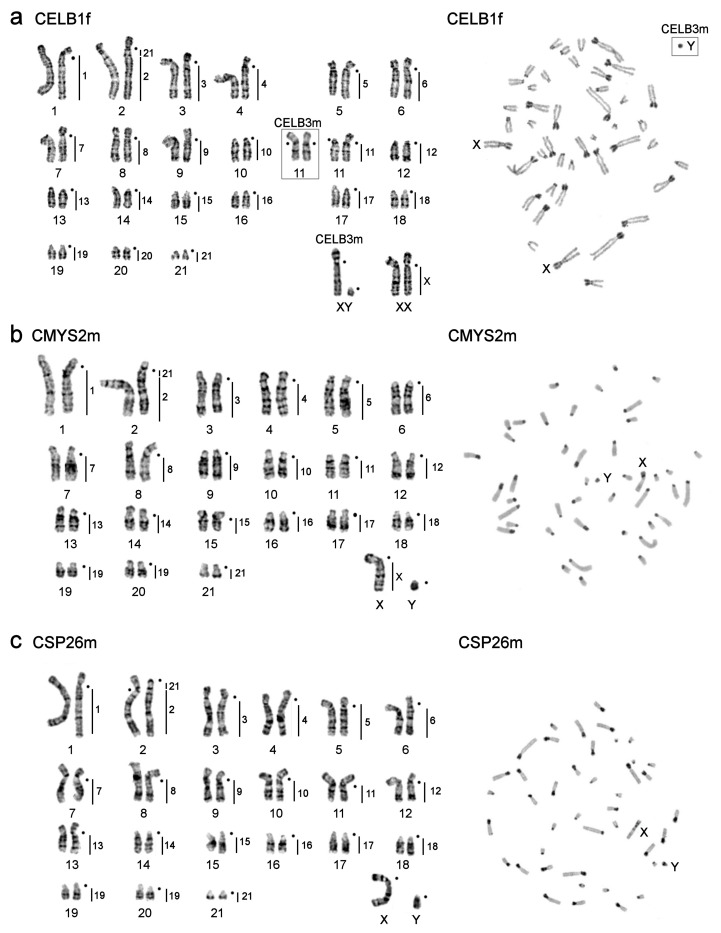
G-banded karyotypes (on the left) and C-banded metaphases (on the right) of the studied specimens with 2n = 44: (**a**) *C. elburzensis* 1f (CELB1f) and *C. elburzensis* 3m (CELB3m); (**b**) *C. m. mystax* 2m (CMYS2m, the karyotype was presented previously in [48]); (**c**) *Calomyscus* sp. 26m (CSP26m)*,* vertical black lines and figures on the right of G-banded chromosomes show localization of *Calomyscus* sp. 17m probes. Black dots mark the position of centromeres. In C-banding photos sex chromosomes (X and Y) are shown.

**Figure 6 genes-12-00964-f006:**
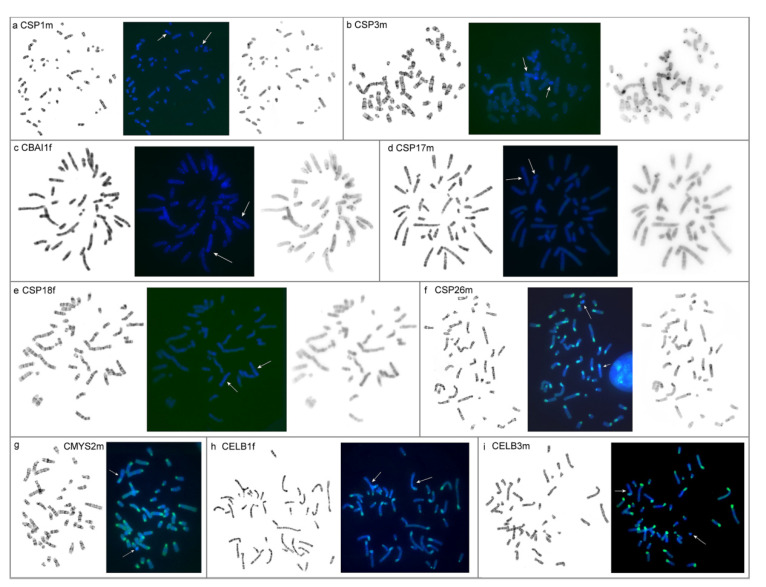
Results of CDAG: (**a**) *Calomyscus* sp. 1m (CSP1m); (**b**) *Calomyscus* sp. 3m (CSP3m); (**c**) *C. bailwardi* 1f (CBAI1f); (**d**) *Calomyscus* sp. 17m (CSP17m); (**e**) *Calomyscus* sp. 18f (CSP18f); (**f**) *Calomyscus* sp. 26m (CSP26m); (**g**) *C. m. mystax* 2m (CMYS2m); (**h**) *C. elburzensis* 1f (CELB1f); (**i**) *C. elburzensis* 3m (CELB3m). From left to right: G-banding, CDAG, inverted DAPI-banding (for some individuals). Blocks of AT-rich heterochromatin (blue) and blocks of GC-rich heterochromatin (green). Sex chromosomes marked by arrows.

**Figure 7 genes-12-00964-f007:**
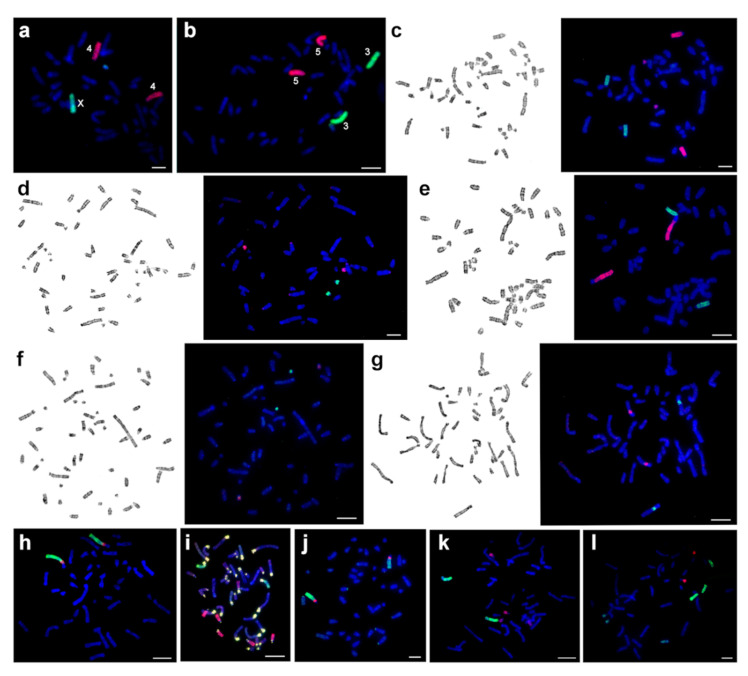
Examples of fluorescent in situ hybridization of chromosome-specific probes onto chromosomes of different *Calomyscus* specimens: (**a**) chromosomes X (green) and 4 (red) of CSP17m onto CSP17m; (**b**) chromosomes 3 (green) and 5 (red) of CSP17m onto CSP17m; (**c**) chromosomes 6 (green) and 4 (red) of CSP17m onto CSP1m; (**d**) microdissection-derived probes C7 (red) and C8 (green) onto CSP1m; (**e**) chromosomes 1 (red) and 6 (green) of CSP17m onto CSP3m; (**f**) microdissection-derived probes C5 (green) and C6 (red) onto CSP3m; (**g**) microdissection-derived probes C7 (red) and C8 (green) onto CELB3m; (**h**) chromosomes 2 (green) and 21 (red) onto CELB1f; (**i**) chromosomes 7 (green) and 11 + 13 + 14 (red) onto CELB1f; (**j**) chromosomes 1 (green) and 21 (red) onto CMYS2m; (**k**) chromosomes 7 (green) and 19 + 20 (red) onto CBAI1f; (**l**) chromosomes 10 + 12 + 19 (green) and 21 (red) onto CSP18f. CBAI—*C. bailwardi,* CELB—*C. elburzensis,* CMYS—*C. m. mystax,* CSP—*Calomyscus* sp. Scale bar is 10 µm.

**Table 1 genes-12-00964-t001:** Origins of the mouse-like hamsters examined in this report.

*Calomyscus* Species, Diploid Chromosome Number (2n)	Geographic Position in Figure 1	Sex	Abbreviation	Locality	Coordinates/Comments
*C. bailwardi* (2n = 46)	1	♀	CBAI1f	Izeh, Khuzestan Province, Iran *	N 31°49′20.8″ E 49°50′09.6″, elevation: 1063 m
*C. elburzensis* (2n = 44)	2	♀	CELB1f	Korkhout Mt., North Khorasan Province, Bojnourd, Iran *	N 37°26′35.2″ E 56°31′30.0″, elevation: 1903 m
*C. elburzensis* (2n = 44)	2	♂	CELB3m	Korkhout Mt.,North Khorasan Province, Bojnourd, Iran *	N 37°26′35.2″ E 56°31′30.0″, elevation: 1903 m (laboratory colony of ZIN RAN)
*C. mystax mystax* (2n = 44)	3	♂	CMYS2m	Maly Balkan Gershi (Arlandag) Mt., Bolshoi Balkhan, Turkmenistan *	N 39°40′20.28″ E 54°32′35.61″, elevation: 1870 m (laboratory colony of ZIN RAN)
*Calomyscus* sp. 1 (2n = 52)	4	♂	CSP1m **	Shahdad tunnel, Kerman Province, Iran	N 30°10′36.41″ E 57°24′44.24″, elevation: 2664 m (laboratory colony of Moscow Zoo)
*Calomyscus* sp. 1 (2n = 52)	4	♂	CSP2m **	Shahdad tunnel, Kerman Province, Iran	N 30°10′36.41″ E 57°24′44.24″, elevation: 2664 m (laboratory colony of Moscow Zoo)
*Calomyscus* sp. 1 (2n = 51)	4	♂	CSP3m **	Shahdad tunnel, Kerman Province, Iran	N 30°10′36.41″ E 57°24′44.24″, elevation: 2664 m (laboratory colony of Moscow Zoo)
*Calomyscus* sp. 2 (2n = 46)	5	♂	CSP17m	Cheshme Maiak, Shahr-e-Kord—Soreshjan, Chahar Mahal and Bakhtiaria Province, Iran	N 32°18′37.9″ E 50°37′34.3″, elevation 2161 m
*Calomyscus* sp. 2 (2n = 46)	5	♀	CSP18f	Aloni, Chahar Mahal and Bakhtiaria Province, Iran	N 31°33′21.3″ E 51°11′25.3″, elevation 1833 m
*Calomyscus* sp. 2 (2n = 46)	5	♂	CSP20m	Cheshme Maiak, Shahr-e-Kord—Soreshjan, Chahar Mahal and Bakhtiaria Province, Iran	N 32°18′37.9″ E 50°37′34.3″, elevation 2161 m
*C. bailwardi* (2n = 46)	5	♂	CSP21m	Izeh, Khozestan Province, Iran	N 31°49′20.8″ E 49°50′09.6″, elevation: 1063 m
*Calomyscus* sp. 2 (2n = 46)	5	♀	CSP22f	Cheshme Maiak, Shahr-e-Kord—Soreshjan, Bakhtiaria Province, Iran	N 32°18′37.9″ E 50°37′34.3″, elevation 2161 m
*Calomyscus* sp. 2 (2n = 46)	5	♀	CSP23f	Aloni, Chahar Mahal and Bakhtiaria Province, Iran	N 31°33′21.3″ E 51°11′25.3″, elevation 1833 m
*Calomyscus* sp. 3 (2n = 44)	6	♂	CSP26m	S-E ridge Zard-Kuh Mts., Isfahan Province, Iran	N 32°27′4.93″ E 50°2′28.36″, elevation 2885 m

* type locality; ** CSP1m, CSP2m, and CSP3m belong to the same form of unclear taxonomic rank.

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
