# Peer review of "New Data on Comparative Cytogenetics of the Mouse-Like Hamsters (Calomyscus Thomas, 1905) from Iran and Turkmenistan"

_genes, 2021, doi:10.3390/genes12070964_

Round 1
Reviewer 1 Report
General comments:
The authors have the merit of using a number of different cytogenetic techniques to address a taxonomic conundrum. The system is fascinating, but I think the authors failed to significantly move forward in the knowledge produced. Considering the amount of intra and interspecific karyotypic variability, the geographic distribution of the different taxa and the overall complexity of the system, the sampling scheme seems inadequate to produce robust results, draw conclusions and significantly increase the knowledge on the taxonomy of the Calomyscus genus. Also, not only the number of analyzed specimens is low and misrepresentative of the complexity and geography of the system, as only 9 of the total 14 were wild caught. Animals originating from lab colonies aiming to represent a single locality exhibited distinct karyotypes among them (2N=51 and 2N=52), raising the questions whether these differences are a result of the breeding process or represent the variability in the wild.
By joining the information here obtained with genome-wide data (see Rawson 2019 -unpublished, to my knowledge), it would result in a more comprehensive understanding of this interesting system.
Specific comments:
Introduction
The Introductions is a bit too long but a thorough review leading to the current knowledge of the system. However, it is difficult to follow with all the different species and detailed locations mentioned and no map to help (the map provided only provides minimal information). I understand this is innate to the system but makes it very difficult for the reader to follow.
Methods
- Considering the variability observed in the karyotypes, very few specimens were analyzed and no robust conclusions can be drawn.
- The methods sub-sections in general are poorly detailed. No protocols are described, only references to previous publications are given. Among them, references to non-open access books and papers are provided. All scientific experiments are supposed to be reproductible and recurrently referring to previous publications non-accessible to others should be avoided (particularly when modifications in protocols occur). If the authors think this would make the methods section too long, further details can be provided as supplementary material.
- Line 127 - Why refer that no human experiments were performed in a study involving hamsters?? Also, considering the date of the permission, it is suggested that the ethical committee’s approval was only after the experiments were carried out and not before.
- Line 130 – Considering that the wild caught specimens have been collected up to 20+ years ago, how can the authors be sure that the consecutive breeding colonies have not accumulated or exhibited occasional chromosomal variation in comparison with their wild ancestors? In fact, among the 3 colony originated specimens, two share the same karyotype but not the other.
- Figure 1 - the figure with the map lacks quality, it is not easy to distinguish the land from the sea. It also lacks a scale. Also, the Zagros mountains and different provinces in Iran are repeatedly mentioned but not shown in the map. Specimens sampled in location 5 and 6 are nearly overlapping in the map provided but from the table analysis, the latitude/longitude show that they are quite far from each other. Again, a more detailed map should be given.
Discussion
The first two paragraphs are redundant with the introduction.
The discussion of the results is limited by the reduced sample size, as are the conclusions to the drawn.
English
The manuscript should be revised by an English native speaker. A couple of examples are listed:
Line 42 – replace “attached to” by “associated with”
Line 60 – delete “as”
Line 62 – delete “a quite”
Line 64 – delete “of time”
Line 72 – delete “of chromosomes”
Line 74 – delete “chromosome”.
Line 89 – replace “variations” for “variation”
Line 98 – replace “marked” for “defined”
Line 105 – replace “karyotype of the forms” for “karyotypic forms”
Line 345 – delete “the”
Author Response
We thank the reviewer for positive view of our paper and for very useful suggestions for improving the presentation.
Reviewer 1
General comments:
The authors have the merit of using a number of different cytogenetic techniques to address a taxonomic conundrum. The system is fascinating, but I think the authors failed to significantly move forward in the knowledge produced. Considering the amount of intra and interspecific karyotypic variability, the geographic distribution of the different taxa and the overall complexity of the system, the sampling scheme seems inadequate to produce robust results, draw conclusions and significantly increase the knowledge on the taxonomy of the Calomyscus genus. Also, not only the number of analyzed specimens is low and misrepresentative of the complexity and geography of the system, as only 9 of the total 14 were wild caught. Animals originating from lab colonies aiming to represent a single locality exhibited distinct karyotypes among them (2N=51 and 2N=52), raising the questions whether these differences are a result of the breeding process or represent the variability in the wild.
We thank the reviewer for careful reading and comments, but we disagree about the lack of novelty. In fact, we have karyotyped natural individuals of all so far described forms and the founder animals from the laboratory colony have been analyzed karyologically and we consistently demonstrated stability of the karyotypes within the colonies, thus they represent a natural complexity.
By joining the information here obtained with genome-wide data (see Rawson 2019 -unpublished, to my knowledge), it would result in a more comprehensive understanding of this interesting system.
Thanks for the info. We carefully read this work, and combined the information with our results.
Specific comments:
Introduction
The Introductions is a bit too long but a thorough review leading to the current knowledge of the system. However, it is difficult to follow with all the different species and detailed locations mentioned and no map to help (the map provided only provides minimal information). I understand this is innate to the system but makes it very difficult for the reader to follow.
We really hope that the changes that we made to Figure 1 (map) will help to better perceive the material.
Methods
- Considering the variability observed in the karyotypes, very few specimens were analyzed and no robust conclusions can be drawn.
We fully agree with this remark, therefore, in our work, we provided only a detailed description of the studied individuals karyotypes, but did not draw any conclusions about the contribution of karyotypic evolution to speciation or about the importance of karyotype description for confirming the status of taxa.
- The methods sub-sections in general are poorly detailed. No protocols are described, only references to previous publications are given. Among them, references to non-open access books and papers are provided. All scientific experiments are supposed to be reproductible and recurrently referring to previous publications non-accessible to others should be avoided (particularly when modifications in protocols occur). If the authors think this would make the methods section too long, further details can be provided as supplementary material.
In line with Genes' requirements, "new methods and protocols should be described in detail, and well-established methods can be briefly described and appropriately cited." All the methods used in this work have been repeatedly described earlier, so we gave only a brief description of them and referred to the primary sources.
- Line 127 - Why refer that no human experiments were performed in a study involving hamsters??
The sentence was deleted.
Also, considering the date of the permission, it is suggested that the ethical committee’s approval was only after the experiments were carried out and not before.
At the meeting of the Ethics Commission of the Zoological Institute, Russian Academy of Sciences, all protocols for working with mouse-like hamsters for all the period of working with them were considered. The project “New Data on Comparative Cytogenetics of the Mouse-like Hamsters (Calomyscus Thomas, 1905) from Iran and Turkmenistan” complies with international and national ethical practices and laws of studies with animals.
- Line 130 – Considering that the wild caught specimens have been collected up to 20+ years ago, how can the authors be sure that the consecutive breeding colonies have not accumulated or exhibited occasional chromosomal variation in comparison with their wild ancestors? In fact, among the 3 colony originated specimens, two share the same karyotype but not the other.
All Turkmen and Iranian forms were first karyotyped by natural individuals (Graphodatsky, A.S.; Sablina, O.V.; Meyer, M.N.; Malikov, V.G.; Isakova, E.A.; Trifonov, V.A.; Polyakov, A.V.; Lushnikova, T.P.; Vorobieva, N.V.; Serdyukova, N.A.; et al. Comparative cytogenetics of hamsters of the genus Calomyscus. Cytogenet. Genome Res. 2000, 88, 296–304, doi:10.1159/000015513). Data on Iranian individuals caught in 2014 (CBAI1f, CELB1f, CSP17 m, and CSP18f) were also previously presented by us (Romanenko SA, Malikov VG, Darvish J., Mahmoudi A., Golenishchev FN. Calomyscus. Is each individual a new species? In: Thesis of International Conference Chromosome 2015. Novosibirsk, p. 151-152 (in russian)) and are presented in this work. In this work, we note that there are some differences between individuals of the same species caught at the same localities at different times, as well as small differences between laboratory individuals. However, we believe that these data do not contradict each other, but only emphasize the high plasticity of the Calomyscus genome and / or currently ongoing speciation process.
- Figure 1 - the figure with the map lacks quality, it is not easy to distinguish the land from the sea. It also lacks a scale. Also, the Zagros mountains and different provinces in Iran are repeatedly mentioned but not shown in the map. Specimens sampled in location 5 and 6 are nearly overlapping in the map provided but from the table analysis, the latitude/longitude show that they are quite far from each other. Again, a more detailed map should be given.
We changed the figure and provided more relevant details.
Discussion
The first two paragraphs are redundant with the introduction.
The discussion of the results is limited by the reduced sample size, as are the conclusions to the drawn.
English
The manuscript should be revised by an English native speaker. A couple of examples are listed:
Line 42 – replace “attached to” by “associated with”
Line 60 – delete “as”
Line 62 – delete “a quite”
Line 64 – delete “of time”
Line 72 – delete “of chromosomes”
Line 74 – delete “chromosome”.
Line 89 – replace “variations” for “variation”
Line 98 – replace “marked” for “defined”
Line 105 – replace “karyotype of the forms” for “karyotypic forms”
Line 345 – delete “the”
We accepted all these changes.

Reviewer 2 Report
In this manuscript, Romanenko et al provided a detailed, careful cytogenetics examination towards 14 Calomyscus species, aiming to provide relevant data to address the long-term controversies regarding the taxonomy of this genus. The new data provided by the authors is informative, the discussions are comprehensive, and the conclusions are drawn with reasonable cautions. However, some data in the current version are not presented clearly, for which the authors should provide additional clarification either in the form of text or additional images. Specific comments are as follow:
- The flow Karyotype data presented in Fig.2 is very difficult to interpret as the peaks are not separated clearly. What are the criteria to identify peaks/chromosomes? The authors should clarify this further in the text. It would be helpful if the authors can provide images for other species as well.
2. The CDAG results (in particular Fig.6) for CSP1m and CSP18f are difficult to judge, the authors should display better quality images.
3. It would be helpful if the authors can provide a schematic presentation at the end of the manuscript to summarise the key karyotype differentiation between different Calomyscus species.
Author Response
We thank the reviewer for positive view of our paper and for very useful suggestions for improving the presentation.
Reviewer 2
Comments and Suggestions for Authors
In this manuscript, Romanenko et al provided a detailed, careful cytogenetics examination towards 14 Calomyscus species, aiming to provide relevant data to address the long-term controversies regarding the taxonomy of this genus. The new data provided by the authors is informative, the discussions are comprehensive, and the conclusions are drawn with reasonable cautions. However, some data in the current version are not presented clearly, for which the authors should provide additional clarification either in the form of text or additional images.
Specific comments are as follow:
- The flow Karyotype data presented in Fig.2 is very difficult to interpret as the peaks are not separated clearly. What are the criteria to identify peaks/chromosomes? The authors should clarify this further in the text. It would be helpful if the authors can provide images for other species as well.
For separation on a flow sorter, chromosome suspension is stained with two fluorochromes (usually chromomycin A3 and Hoechst are used). When passing through laser beams, chromosomes are separated into separate tubes depending on their size and GC-content. With the help of software, the intensity of fluorescence of each chromosome is determined, quantitative information about the sorted chromosomes is summarized and displayed in the form of a graphic image - flow-karyotype. Dark areas indicate peak positions. Chromosomes of similar size and GC-content may end up in one tube (one peak). To determine the content of each peak, the amplified and labeled DNA of the sorted chromosomes is localized by FISH on the metaphase chromosomes of the parent species.
In our work, the chromosomes of only one individual were sorted. The obtained DNA libraries were used as probes for comparing the karyotypes of all individuals analyzed in the work.
- The CDAG results (in particular Fig.6) for CSP1m and CSP18f are difficult to judge, the authors should display better quality images.
The work contains images of the best quality. Since the specimens CSP1m and CSP18m do not contain large blocks of AT- and GC-rich heterochromatin, CDAG does not give a clear picture.
- It would be helpful if the authors can provide a schematic presentation at the end of the manuscript to summarise the key karyotype differentiation between different Calomyscus species.
We opted out of the graphical presentation of the results due to the difficulty of describing small differences between species caused by variations in the amount and distribution of heterochromatin. It seems to us that at the current stage such work will be largely speculative.

Reviewer 3 Report
Your manuscript is informative for the consideration of taxonomy in the mouse-like hamsters. However, some technical term usages for taxonomic explanations are wrong.
My comments and questions are following,
L45: What do “comparative molecular-genetic analyses” mean? Clarify here in detail.
L46: What does “monotypic” mean? Morphologically or phylogenetically? If the former, is “brush-tail” specific to this family? The current expression of this sentence is not understandable. Clarify this sentence.
L63: As in above, what does “monotypic” mean? Although you used “monotypic” throughout this manuscript, primarily you should define “monotype.”
L63: In this manuscript, you frequently use “nominal form.” What does “nominal” mean? Does “nominal form” mean species and/or subspecies? In Introduction, you introduced chronological taxonomic histories of Calomyscus, but some usages of words are not appropriate for taxonomic explanations, as pointed out above.
L92: What does “Specific status” mean? Specific validity? Clarify this sentence such as “C. urartensis and C. mystax were recognized as different species.”
L105–110: This sentence is too long!
L118: 14 Calomyscus specimens > 14 specimens of Calomyscus spp.
L184: Do not italic “sp.” You should check entirely in text.
L188: You mentioned “a small interstitial heterochromatic block on acrocentric X chromosome.” However, in Fig. 3, the C-bands on X seemed to be not interstitial but terminal. By the way, Figs. 3, 4 and 5 photos were quite small and it is difficult to recognize C-bands. Indicate more larger photos of C-banding in these Figs.
L193 and others: Better to use short-arm than p-arm. It is kind for beginners of cytogenetics.
L193 and others: Better to use long-arm than q-arm.
L234–240: Photos of Fig. 6 were quite small! I cannot see staining patterns of comparisons. Indicate more larger photos.
L298: “identification” was wrong. Here, you should use “recognition” or “classification.” Do you know meaning of identification? Your usage of some terms related to taxonomy was wrong.
L307: populated by > colonized by or consisted of
L321: karyotyped > karyotypes?
L348: What does “In CELB1f the X chromosomes have clearer C-banding patterns” mean? Is this sentence necessary? After here, you did not mention about the X chromosomes.
L349: I cannot understand “The obvious disagreement in chromosome morphology...” What does this sentence mean? Clarify here.
L353: Do you know the difference between “species name” and ”specific name?” Species name = generic name + specific name and your usage was wrong. If you use “specific name,” “elburzensis” only (without generic name) is correct. Here, “The similarity of C. elburzensis type karyotype to karyotype 3 should show that the specific name C. firiuzaensis is a synonym of C. elburzensis.” should be as “Considering the similarity of C. elburzensis type karyotype to karyotype 3, C. firiuzaensis should be a junior synonym of C. elburzensis taxonomically.”
L359: What does “heterochromatin change” mean? Quantitative variation of heterochromatin? Heterochromatinization? Clarify here.
L363: C. m. mystax, but in L106, C. mystax mystax. Unify trinominal usages throughout manuscript.
L367: What does “description” mean? This sentence should be as “The description of C. m. mystax karyotype indicates a correspondence to the karyotype of ...” or “The karyotype of C. m. mystax corresponds to that of ...”
L374: Here, you should briefly introduce chromosomal features, for example 2n numbers, of A. maximowiczii group.
Fig. 1: Do not italic “sp.”
Fig. 2 caption: There is “*see comment in the text.” Where do readers see? L255? This caption is quite unkind.
Figs. 3, 4 and 5 captions: Which is C-banding column? Indicate G- and C- banding profiles.
Figs. 3, 4 and 5 captions: “G- and C-banded karyotypes” were wrong. “G-banded karyotypes and C-banded metaphases” were correct. Dou you understand meaning of “karyotype”?
Figs. 3, 4 and 5 captions: Indications were scarce in C-banding photos. For example, in Fig. 3, indicate interstitial (terminal?) heterochromatins and in Fig. 4b, indicate a heterochromatin segment in the largest chromosome.
Fig. 3 caption: Do not italic “sp.” (L196)
Your English expressions were sometimes not general (sometimes difficult to understand) and English qualities were partially inadequate. Check English by a native speaker.
Author Response
We thank the reviewer for positive view of our paper and for very useful suggestions for improving the presentation.
Reviewer 3
Comments and Suggestions for Authors
Your manuscript is informative for the consideration of taxonomy in the mouse-like hamsters. However, some technical term usages for taxonomic explanations are wrong.
My comments and questions are following,
L45: What do “comparative molecular-genetic analyses” mean? Clarify here in detail.
We apologize for providing an incorrect reference. In this case, we cited work by Michaux J, Reyes A, Catzeflis F (2001) Evolutionary history of the most speciose mammals: molecular phylogeny of muroid rodents. Mol Biol Evol 18:2017–2031.
L46: What does “monotypic” mean? Morphologically or phylogenetically? If the former, is “brush-tail” specific to this family? The current expression of this sentence is not understandable. Clarify this sentence.
The term "monotypic" refers to taxonomy and it is commonly used. It means that this taxonomic category is not subdivided into categories that are lower in the taxonomic hierarchy. That is, monotype is a nomenclature situation in which only one taxon of a lower rank belongs to a taxon of a higher rank. So, in a monotypic genus there is only one species, and in a monotypic family includes only one genus.
L63: As in above, what does “monotypic” mean? Although you used “monotypic” throughout this manuscript, primarily you should define “monotype.”
Please, see the previous comment.
L63: In this manuscript, you frequently use “nominal form.” What does “nominal” mean? Does “nominal form” mean species and/or subspecies? In Introduction, you introduced chronological taxonomic histories of Calomyscus, but some usages of words are not appropriate for taxonomic explanations, as pointed out above.
A “nominal form” is any form that has ever been described as a separate nomenclatural unit, regardless of its validity and taxonomic rank.
L92: What does “Specific status” mean? Specific validity? Clarify this sentence such as “C. urartensis and C. mystax were recognized as different species.”
We meant «species». The sentence was corrected following your suggestion.
L105–110: This sentence is too long!
The sentence was split on two.
L118: 14 Calomyscus specimens > 14 specimens of Calomyscus spp.
Suggested correction was made.
L184: Do not italic “sp.” You should check entirely in text.
There were only two italic “sp.” In the text. In the both cases text was corrected.
L188: You mentioned “a small interstitial heterochromatic block on acrocentric X chromosome.” However, in Fig. 3, the C-bands on X seemed to be not interstitial but terminal. By the way, Figs. 3, 4 and 5 photos were quite small and it is difficult to recognize C-bands. Indicate more larger photos of C-banding in these Figs.
In all cases, the interstitial block of heterochrmatin is located in the distal, non-terminal region of the X chromosome.
We chose this type of presentation of illustrations in order to reduce their total number in manuscript and not to transfer figures to the supplement. In our opinion, the presentation of the C-banding results in separate figures will only complicate the perception of the material.
L193 and others: Better to use short-arm than p-arm. It is kind for beginners of cytogenetics.
L193 and others: Better to use long-arm than q-arm.
We have added a clarification to the text that p-arm is short and q-arm is long.
L234–240: Photos of Fig. 6 were quite small! I cannot see staining patterns of comparisons. Indicate more larger photos.
Our work contains a detailed description of the karyotypes of 14 individuals. We would very much like the main data to be presented as much as possible in the text, and not transferred to a supplement. That is why we chose this form of presentation of the results.
L298: “identification” was wrong. Here, you should use “recognition” or “classification.” Do you know meaning of identification? Your usage of some terms related to taxonomy was wrong.
We chanced “identification” to “recognition”.
L307: populated by > colonized by or consisted of
Replaced by “colonized by”.
L321: karyotyped > karyotypes?
The sentence was corrected: “It is possible that the differences between the C. bailwardi karyotypes described here and presented by Radjabli et al. [49] can be explained by incorrect species identification in the previously published case.”
L348: What does “In CELB1f the X chromosomes have clearer C-banding patterns” mean? Is this sentence necessary? After here, you did not mention about the X chromosomes.
The sentence was removed.
L349: I cannot understand “The obvious disagreement in chromosome morphology...” What does this sentence mean? Clarify here.
We clarified in the text that in the work of Radjabli et al. [49], the pair 7 is acrocentric on G-banded and submetacentric in C-banded karyotypes.
L353: Do you know the difference between “species name” and ”specific name?” Species name = generic name + specific name and your usage was wrong. If you use “specific name,” “elburzensis” only (without generic name) is correct. Here, “The similarity of C. elburzensis type karyotype to karyotype 3 should show that the specific name C. firiuzaensis is a synonym of C. elburzensis.” should be as “Considering the similarity of C. elburzensis type karyotype to karyotype 3, C. firiuzaensis should be a junior synonym of C. elburzensis taxonomically.”
Indeed, C. firiuzaensis should be considered a junior synonym for C. elburzensis. We considered your comment and corrected the text.
L359: What does “heterochromatin change” mean? Quantitative variation of heterochromatin? Heterochromatinization? Clarify here.
We meant “variations”. The sentence was corrected.
L363: C. m. mystax, but in L106, C. mystax mystax. Unify trinominal usages throughout manuscript.
Done.
L367: What does “description” mean? This sentence should be as “The description of C. m. mystax karyotype indicates a correspondence to the karyotype of ...” or “The karyotype of C. m. mystax corresponds to that of ...”
The sentence was corrected following your suggestion.
L374: Here, you should briefly introduce chromosomal features, for example 2n numbers, of A. maximowiczii group.
We do not consider it necessary to provide clarifying information on “maximowiczii”-group here, since it is not directly related to our work.
Fig. 1: Do not italic “sp.”
The Figure was corrected.
Fig. 2 caption: There is “*see comment in the text.” Where do readers see? L255? This caption is quite unkind.
We deleted the reference to the text and changed the figure caption to “Bivariate flow karyotype of Calomyscus sp. (CSP17m) cell line with chromosomal assignments.”
Figs. 3, 4 and 5 captions: Which is C-banding column? Indicate G- and C- banding profiles.
We changed the figure capture and clarified that the karyotype is shown on the left and the metaphase is shown on the right. We decided not to add a profile so as not to overload the figure.
Figs. 3, 4 and 5 captions: “G- and C-banded karyotypes” were wrong. “G-banded karyotypes and C-banded metaphases” were correct. Dou you understand meaning of “karyotype”?
We corrected the figures captures following your suggestion.
Figs. 3, 4 and 5 captions: Indications were scarce in C-banding photos. For example, in Fig. 3, indicate interstitial (terminal?) heterochromatins and in Fig. 4b, indicate a heterochromatin segment in the largest chromosome.
We expanded the captions for these figures.
Fig. 3 caption: Do not italic “sp.” (L196)
Corrected.
Your English expressions were sometimes not general (sometimes difficult to understand) and English qualities were partially inadequate. Check English by a native speaker.
English was checked and corrected.

Round 2
Reviewer 1 Report
I'll be happy with the outcome when the authors address the following concerns:
Authors reply- We thank the reviewer for careful reading and comments, but we disagree about the lack of novelty. In fact, we have karyotyped natural individuals of all so far described forms and the founder animals from the laboratory colony have been analyzed karyologically and we consistently demonstrated stability of the karyotypes within the colonies, thus they represent a natural complexity.
Reply Reviewer 1 - If this is this case, then it should be stated in the manuscript (in section 2.2. Species Sampled).
Authors reply - In line with Genes' requirements, "new methods and protocols should be described in detail, and well-established methods can be briefly described and appropriately cited." All the methods used in this work have been repeatedly described earlier, so we gave only a brief description of them and referred to the primary sources.
Reply Reviewer 1 - The authors did not provide any brief descriptions of any of the techniques, only referring to previous work. Again, at least, cited literature should provide readers the possibility to find protocols in open access. Being a very descriptive paper, at least it should provide its reader with
Author Response
Reviewer 1
I'll be happy with the outcome when the authors address the following
concerns:
Authors reply- We thank the reviewer for careful reading and comments,
but we disagree about the lack of novelty. In fact, we have karyotyped
natural individuals of all so far described forms and the founder
animals from the laboratory colony have been analyzed karyologically and
we consistently demonstrated stability of the karyotypes within the
colonies, thus they represent a natural complexity.
Reply Reviewer 1 - If this is this case, then it should be stated in the
manuscript (in section 2.2. Species Sampled).
The following sentence was added to the text: “It must be emphasized that in fact, we karyotyped natural individuals of all forms described so far, and the founder animals from the laboratory colonies were also analyzed karyologically. We consistently demonstrated stability of the karyotypes within the colonies, thus they represent a natural complexity.”
Authors reply - In line with Genes' requirements, "new methods and
protocols should be described in detail, and well-established methods
can be briefly described and appropriately cited." All the methods used
in this work have been repeatedly described earlier, so we gave only a
brief description of them and referred to the primary sources.
Reply Reviewer 1 - The authors did not provide any brief descriptions of
any of the techniques, only referring to previous work. Again, at least,
cited literature should provide readers the possibility to find
protocols in open access. Being a very descriptive paper, at least it
should provide its reader with
Unfortunately, the comment looks incomplete. We included very short description for techniques used.

Reviewer 3 Report
L99: Your word usage ‘distinct monotypic family’ was not appropriate. What can you distinct from?
the genus was considered to belong to the distinct monotypic family of brush-tailed mice, Calomyscidae Vorontsov and Potapova, 1979 [8]. ==> the genus was considered as a taxon of a monotypic family, Calomyscidae Vorontsov and Potapova, 1979, characterized by brush-tailed appearance [8].
L121–: You used ‘nominal form’ but you should explain without ‘nominal form’…
> It is unclear which does ‘nominal form’ in L131 mean species or subspecies.
> Meaning of ‘its nominal forms as subspecies’ is still unclear. Only susbspecies?
Entirely in this manuscript, specific and subspecific ranks were appeared and the taxonomic status has been complicated. Your usage of ‘nominal form’ is unkind for readers because your explanation included both specific and subspecific ranks, leading complication. Therefore, you should mention each taxonomic status without ‘nominal form.’
L418 and C-banded metaphases: Indicate interstitial heterochromatin on X, using arrowheads or other markings! These photos were too small and as in my previous comments, your indications were scarce. By the way, in your previous comment, “In all cases, the interstitial block of heterochromatin is located in the distal, non-terminal region of the X chromosome.” What does this sentence mean? Is the interstitial block located in the distal region?
L730: You answered as “We do not consider it necessary to provide clarifying information on “maximowiczii” group here, since it is not directly related to our work.” If so, why did you refer this taxon?
Incidentally, What are species as members of the “maximowiczii” group? In the group, what kind of heteromorphisms? Robertsonian variations? Other rearrangements? If you refer these voles, you should introduce typical examples, species and its heteromorphic types, related to your findings.
Author Response
Reviewer 3
L99: Your word usage ‘distinct monotypic family’ was not appropriate.
What can you distinct from?
We did not find the phrase ‘distinct monotypic family’ in the text.
the genus was considered to belong to the distinct monotypic family of
brush-tailed mice, Calomyscidae Vorontsov and Potapova, 1979 [8]. ==>
the genus was considered as a taxon of a monotypic family, Calomyscidae
Vorontsov and Potapova, 1979, characterized by brush-tailed appearance
[8].
The sentence was corrected following your suggestion.
L121–: You used ‘nominal form’ but you should explain without ‘nominal
form’…
It is unclear which does ‘nominal form’ in L131 mean species or subspecies.
Meaning of ‘its nominal forms as subspecies’ is still unclear. Only susbspecies?
Entirely in this manuscript, specific and subspecific ranks were
appeared and the taxonomic status has been complicated. Your usage of
‘nominal form’ is unkind for readers because your explanation included
both specific and subspecific ranks, leading complication. Therefore,
you should mention each taxonomic status without ‘nominal form.’
We abandoned the use of this term in the text (lines 58, 64, 70, and 94).
L418 and C-banded metaphases: Indicate interstitial heterochromatin on
X, using arrowheads or other markings! These photos were too small and
as in my previous comments, your indications were scarce. By the way, in
your previous comment, “In all cases, the interstitial block of
heterochromatin is located in the distal, non-terminal region of the X
chromosome.” What does this sentence mean? Is the interstitial block
located in the distal region?
Yes, we meant that the interstitial block located in the distal region of acrocentric X chromosomes. This clarification was added to the text. We marked the interstitial C-positive block on X in Figure 3.
L730: You answered as “We do not consider it necessary to provide
clarifying information on “maximowiczii” group here, since it is not
directly related to our work.” If so, why did you refer this taxon?
We wanted to emphasize that the cytogenetic variations observed in the genus Calomyscus are not unique, but are found in other groups of rodents. We removed the mention of the “maximowiczii” group from the text.
Incidentally, What are species as members of the “maximowiczii” group?
In the group, what kind of heteromorphisms? Robertsonian variations?
Other rearrangements? If you refer these voles, you should introduce
typical examples, species and its heteromorphic types, related to your
findings.
The group “maximowiczii” (Arvicolinae, Alexandromys) includes three closely related species of the Far East - A. mujanensis, A. mujanensis, and A. evoronensis. All three species are characterized by a high level of chromosomal polymorphism - percentric and paracentric inversions, centromere repositions, interchromosomal rearrangements (centric, telomeric, and centromeric-telomeric fusions), and heterochromatin variations. We removed the mention of the “maximowiczii” group in the text.
